# Patterns of COVID-19 testing and mortality by race and ethnicity among United States veterans: A nationwide cohort study

Christopher T. Rentsch[1,2]*, Farah Kidwai-Khan[1,3], Janet P. Tate[1,3], Lesley S. Park[4], Joseph T. King Jr[1,5], Melissa Skanderson[1], Ronald G. Hauser[1,6], Anna Schultze[2], Christopher I. Jarvis[2], Mark Holodniy[7,8], Vincent Lo Re III[9,10], Kathleen M. Akgün[1,3], Kristina Crothers[11,12], Tamar H. Taddei[1,3], Matthew S. Freiberg[13,14], Amy C. Justice[1,3,15]

1 VA Connecticut Healthcare System, US Department of Veterans Affairs, West Haven, Connecticut, United States of America, 2 Faculty of Epidemiology and Population Health, London School of Hygiene & Tropical Medicine, London, United Kingdom, 3 Department of Internal Medicine, Yale School of Medicine, New Haven, Connecticut, United States of America, 4 Stanford Center for Population Health Sciences, Stanford University School of Medicine, Stanford, California, United States of America, 5 Department of Neurosurgery, Yale School of Medicine, New Haven, Connecticut, United States of America, 6 Department of Laboratory Medicine, Yale School of Medicine, New Haven, Connecticut, United States of America, 7 VA Palo Alto Health Care System, US Department of Veterans Affairs, Palo Alto, California, United States of America, 8 Division of Infectious Diseases and Geographic Medicine, Stanford University School of Medicine, Stanford, California, United States of America, 9 Division of Infectious Diseases, Department of Medicine, Perelman School of Medicine, University of Pennsylvania, Philadelphia, Pennsylvania, United States of America, 10 Center for Clinical Epidemiology and Biostatistics, Department of Biostatistics, Epidemiology, and Informatics, Perelman School of Medicine, University of Pennsylvania, Philadelphia, Pennsylvania, United States of America, 11 VA Puget Sound Health Care System, US Department of Veterans Affairs, Seattle, Washington, United States of America, 12 Department of Medicine, University of Washington School of Medicine, Seattle, Washington, United States of America, 13 Geriatric Research Education and Clinical Center, Tennessee Valley Healthcare System, US Department of Veterans Affairs, Nashville, Tennessee, United States of America, 14 Department of Medicine, Vanderbilt University Medical Center, Nashville, Tennessee, United States of America, 15 Center for Interdisciplinary Research on AIDS, Yale School of Public Health, New Haven, Connecticut, United States of America

☯ These authors contributed equally to this work.
* christopher.rentsch@lshtm.ac.uk

## Abstract

### Background

There is growing concern that racial and ethnic minority communities around the world are experiencing a disproportionate burden of severe acute respiratory syndrome coronavirus 2 (SARS-CoV-2) infection and coronavirus disease 2019 (COVID-19). We investigated racial and ethnic disparities in patterns of COVID-19 testing (i.e., who received testing and who tested positive) and subsequent mortality in the largest integrated healthcare system in the United States.

### Methods and findings

This retrospective cohort study included 5,834,543 individuals receiving care in the US Department of Veterans Affairs; most (91%) were men, 74% were non-Hispanic White (White), 19% were non-Hispanic Black (Black), and 7% were Hispanic. We evaluated

**Data Availability Statement:** Due to US Department of Veterans Affairs (VA) regulations and our ethics agreements, the analytic data sets

used for this study are not permitted to leave the VA firewall without a Data Use Agreement. This limitation is consistent with other studies based on VA data. However, VA data are made freely available to researchers with an approved VA study protocol. For more information, please visit https://www.virec.research.va.gov or contact the VA Information Resource Center at VIReC@va.gov.

**Funding:** This work was supported by the National Institute on Alcohol Abuse and Alcoholism [ACJ: U01-AA026224, U24-AA020794, U01-AA020790, U10-AA013566]. The funders had no role in study design, data collection and analysis, decision to publish, or preparation of the manuscript.

**Competing interests:** The authors have declared that no competing interests exist.

**Abbreviations:** COVID-19, coronavirus disease 2019; OR, odds ratio; SARS-CoV-2, severe acute respiratory syndrome coronavirus 2; VA, Department of Veterans Affairs.

associations between race/ethnicity and receipt of COVID-19 testing, a positive test result, and 30-day mortality, with multivariable adjustment for a wide range of demographic and clinical characteristics including comorbid conditions, health behaviors, medication history, site of care, and urban versus rural residence. Between February 8 and July 22, 2020, 254,595 individuals were tested for COVID-19, of whom 16,317 tested positive and 1,057 died. Black individuals were more likely to be tested (rate per 1,000 individuals: 60.0, 95% CI 59.6–60.5) than Hispanic (52.7, 95% CI 52.1–53.4) and White individuals (38.6, 95% CI 38.4–38.7). While individuals from minority backgrounds were more likely to test positive (Black versus White: odds ratio [OR] 1.93, 95% CI 1.85–2.01, $p < 0.001$; Hispanic versus White: OR 1.84, 95% CI 1.74–1.94, $p < 0.001$), 30-day mortality did not differ by race/ethnicity (Black versus White: OR 0.97, 95% CI 0.80–1.17, $p = 0.74$; Hispanic versus White: OR 0.99, 95% CI 0.73–1.34, $p = 0.94$). The disparity between Black and White individuals in testing positive for COVID-19 was stronger in the Midwest (OR 2.66, 95% CI 2.41–2.95, $p < 0.001$) than the West (OR 1.24, 95% CI 1.11–1.39, $p < 0.001$). The disparity in testing positive for COVID-19 between Hispanic and White individuals was consistent across region, calendar time, and outbreak pattern. Study limitations include underrepresentation of women and a lack of detailed information on social determinants of health.

## Conclusions

In this nationwide study, we found that Black and Hispanic individuals are experiencing an excess burden of SARS-CoV-2 infection not entirely explained by underlying medical conditions or where they live or receive care. There is an urgent need to proactively tailor strategies to contain and prevent further outbreaks in racial and ethnic minority communities.

## Author summary

### Why was this study done?

- There is growing concern that racial and ethnic minority communities around the world are experiencing a disproportionate burden of morbidity and mortality from symptomatic severe acute respiratory syndrome coronavirus 2 (SARS-CoV-2) infection or coronavirus disease 2019 (COVID-19).

- Most studies investigating racial and ethnic disparities to date have focused on those who tested positive for SARS-CoV-2 or hospitalized patients.

- No single study to our knowledge has yet investigated racial and ethnic disparities in testing patterns (i.e., who received testing and who tested positive) as well as COVID-19 outcomes in a nationwide cohort with adequate adjustment for potential confounders.

### What did the researchers do and find?

- We used electronic health records from the largest integrated healthcare system in the US to investigate racial and ethnic disparities in testing and subsequent COVID-19 mortality.

- Non-Hispanic Black and Hispanic individuals were twice as likely as non-Hispanic White individuals to test positive for COVID-19, even after accounting for other demographics, geographic location, and underlying health conditions.

- The racial disparity between Black and White individuals in testing positive for COVID-19 slightly decreased over the study period, and was highest in the Midwest compared to all other regions. The ethnic disparity between Hispanic and White individuals in testing positive for COVID-19 was consistent across time, geographic region, and outbreak pattern; the disparity was consistently observed across all strata.

- Among those who tested positive for COVID-19, there was no observed difference in 30-day mortality by race/ethnicity group.

### What do these findings mean?

- Our findings highlight the urgent need for improved strategies to contain and prevent further outbreaks in racial and ethnic minority communities.

## Introduction

The United States has the highest number of reported symptomatic severe acute respiratory syndrome coronavirus 2 (SARS-CoV-2) infections and related deaths in the world, accounting for one-fourth of global totals as of July 22, 2020 [1]. There is growing concern that racial and ethnic minority communities are experiencing a disproportionate burden of morbidity and mortality from symptomatic SARS-CoV-2 infection or coronavirus disease 2019 (COVID-19) [2–8]. One statewide study investigating racial disparities followed 3,481 COVID-19 cases in Louisiana and found that non-Hispanic Black individuals represented 77% of hospitalizations and 71% of deaths despite only making up 31% of the total source population [9]. Thus, the potential for racial and ethnic disparities in COVID-19 have been deemed an urgent public health research priority [10]. However, most studies investigating racial and ethnic disparities have focused on hospitalized patients or have not characterized who received testing or tested positive for COVID-19 [9,11–15]. Given that COVID-19 testing was not performed at random, particularly in the early phases of the pandemic, evaluating underlying testing patterns and changes over time may provide important context for interpreting findings from models of COVID-19 outcomes. In addition, it is not yet known whether disparities in COVID-19 infection or severe outcomes are explained, at least in part, by differences in underlying health conditions, smoking and alcohol use, geographic location, or urban versus rural residence—essential information if we are to design effective interventions.

The electronic health record database of the Department of Veterans Affairs (VA) offers the single largest nationwide data resource available with the necessary information on system-wide testing and detailed medical histories to examine racial and ethnic disparities in the US. We evaluated associations between race/ethnicity and receipt of COVID-19 testing, a positive test result, and 30-day mortality, conditioning each analysis on the previous outcome and accounting for a wide range of demographic and clinical characteristics through July 22, 2020.

## Methods

### Data source

The VA is the largest integrated healthcare system in the US and comprises over 1,200 points of care (i.e., sites) nationwide including hospitals, medical centers, and community outpatient clinics. All care is recorded in an electronic health record with daily uploads into the VA Corporate Data Warehouse. Available data include demographics, outpatient and inpatient encounters, diagnoses, smoking and alcohol health behaviors, and pharmacy dispensing records.

This study was approved by the institutional review boards of VA Connecticut Healthcare System and Yale University. It has been granted a waiver of informed consent and is Health Insurance Portability and Accountability Act compliant. The analyses herein were not prespecified in a formal protocol, rather were informed by hypotheses drawn from prior work [16]. This study is reported as per the Strengthening the Reporting of Observational Studies in Epidemiology (STROBE) guidelines (S1 STROBE Checklist).

### Sample, follow-up, and outcomes

All individuals in clinical care (defined as having at least 1 clinical encounter between January 1, 2018, and December 31, 2019, and alive as of January 1, 2020) were included in this analysis. We identified individuals tested for COVID-19 from the date of the VA's first recorded test, on February 8, 2020, through July 22, 2020, by using text searching of laboratory results containing terms consistent with SARS-CoV-2 or COVID-19. Nearly all tests utilized nasopharyngeal swabs; 1% were from other sources. Testing was performed in VA, state public health, and commercial reference laboratories using FDA Emergency Use Authorization–approved SARS-CoV-2 assays. We did not include antibody tests in this analysis.

If an individual had more than 1 test and all were negative, we selected the date of the first negative test; otherwise we used the date of the first positive test. Baseline for individuals tested for COVID-19 was defined as the date of specimen collection unless testing occurred during hospitalization, in which case baseline was defined as the date of admission. If the admission began more than 14 days prior to testing, which may indicate hospital-acquired infection, we set baseline to 14 days prior to testing to better capture health status prior to SARS-CoV-2 infection. We examined 3 outcomes: (1) receipt of COVID-19 testing among all in care, (2) receipt of a positive test result among individuals tested for COVID-19, and (3) 30-day mortality among COVID-19 cases. Deaths were ascertained using inpatient records and VA death registry data to capture deaths outside of hospitalization. The choice of 30-day mortality as the outcome was guided by the distribution of mortality events by time since testing positive for COVID-19 (50th, 75th, 90th percentile time to death: 12, 20, 30 days) and to allow for sufficient follow-up within the study period. While there were some deaths beyond 30 days after testing positive for COVID-19, we were less certain that these deaths could be attributed to COVID-19. Given the low number of deaths after 30 days, 30-day mortality may be a reasonable proxy for case fatality rate. However, until longer follow-up has accrued, it remains to be seen whether those who develop symptomatic COVID-19 experience longer term excess mortality.

### Variables

The primary exposure variable was self-reported race/ethnicity (non-Hispanic White [White], non-Hispanic Black [Black], and Hispanic). Analyses of other racial and ethnic backgrounds were underpowered at the time of this analysis, and therefore individuals who self-reported race/ethnicity other than White, Black, or Hispanic were excluded from the study population.

We selected demographic and clinical characteristics that have been evaluated in prior COVID-19 reports and could potentially mediate or explain racial/ethnic disparities in COVID-19 positivity and mortality. Demographics included age at baseline, sex, and rural/urban residence. Rural/urban residence was defined using geographic information system coding based upon established criteria [17]. Clinical characteristics were based on diagnostic codes for asthma, any cancer, chronic obstructive pulmonary disease (COPD), chronic kidney disease, diabetes mellitus, hypertension, liver disease, vascular disease, and alcohol use disorder (definitions provided in S1 Table). Presence of conditions was determined by 1 inpatient or 2 outpatient diagnoses in the 2 years prior to baseline, except for cancer, which was considered present if diagnosed ever prior to baseline. Diagnoses made in the 7 days prior to baseline were not included. We used a validated algorithm to capture smoking status [18] and alcohol consumption [19]. We collected pharmacy fills for angiotensin converting enzyme (ACE) inhibitors and angiotensin II receptor blockers (ARBs) and identified individuals with active prescriptions in the 30 days prior to baseline. Missing data for smoking and alcohol consumption affected only 5% of individuals included in multivariable models; thus, complete case analysis was performed.

We also created variables to assess potential variation in racial/ethnic disparities by calendar time, region, and outbreak pattern. We split the population into 3 groups based on date of COVID-19 test: February 8 to April 21, April 22 to June 21, and June 22 to July 22. States were grouped into 4 US Census regions (i.e., West, South, Midwest, and Northeast) [20]. Outbreak patterns were based on site-level percentage of positive tests per month among sites with at least 100 positive COVID-19 tests: early (≥10% in March or April), late (≥10% in June or July), resurgent (≥10% in March or April and June or July), steady (<10% in all months), and other (sites with <100 positive tests).

## Statistical analysis

We calculated COVID-19 testing rate per 1,000 individuals in care and Clopper–Pearson 95% confidence intervals (CIs) by race/ethnicity category. Among those tested for COVID-19, we calculated percent testing positive and 95% CIs by race/ethnicity category. Logistic regression models were used to estimate associations between race/ethnicity and COVID-19 positivity and mortality, adjusting for sets of potential mediators of such disparities, moving from more distal to more proximate determinants of health. Age-adjusted models included race/ethnicity and age. Demographic-adjusted models additionally included sex and rural/urban residence, and were conditioned on site of care. Fully adjusted models additionally included all clinical covariates, substance use, and medication history. We report the estimates of each individual adjustment as well as those arising from a fully adjusted model. We repeated this modeling strategy to estimate odds ratios (ORs) and 95% CIs between race/ethnicity and 30-day mortality among those who tested positive for COVID-19 on or prior to June 21, 2020, to allow all individuals 30 days of follow-up. We evaluated variation in racial/ethnic disparities in testing positive for COVID-19 by stratifying the fully adjusted model by calendar time, geographic region, and site-level outbreak pattern. In sensitivity analyses, we restricted ascertainment of 30-day mortality to only include inpatient deaths to test the robustness of the associations found in the primary models. Analyses were performed using SAS version 9.4 (SAS Institute, Cary, NC, US). R version 3.6.3 was used to map COVID-19 cases nationwide.

## Results

There were 5,834,543 individuals in care prior to the COVID-19 pandemic. Most (91%) were men, 74% were White, 19% were Black, and 7% were Hispanic (Table 1). Age distributions

**Table 1. Characteristics of all individuals who were in care, were tested, tested positive, and died as of July 22, 2020.**

| Characteristic | In care | | Tested | | Tested positive | | Died | |
|---|---|---|---|---|---|---|---|---|
| | Number | Column percent | Number | Column percent | Number | Column percent | Number | Column percent |
| Sample size, *n* (%) | 5,834,543 | 100.0% | 254,595 | 100.0% | 16,317 | 100.0% | 1,057 | 100.0% |
| Race/ethnicity | | | | | | | | |
| White | 4,309,613 | 73.9% | 166,213 | 65.3% | 7,159 | 43.9% | 525 | 49.7% |
| Black | 1,089,883 | 18.7% | 65,441 | 25.7% | 6,589 | 40.4% | 433 | 41.0% |
| Hispanic | 435,047 | 7.5% | 22,941 | 9.0% | 2,569 | 15.7% | 99 | 9.4% |
| Age, years | | | | | | | | |
| 20–39 | 842,948 | 14.4% | 28,695 | 11.3% | 2,719 | 16.7% | 1 | 0.1% |
| 40–49 | 577,135 | 9.9% | 23,422 | 9.2% | 1,879 | 11.5% | 11 | 1.0% |
| 50–59 | 840,779 | 14.4% | 42,877 | 16.8% | 3,071 | 18.8% | 52 | 4.9% |
| 60–69 | 1,210,960 | 20.8% | 65,308 | 25.7% | 3,705 | 22.7% | 192 | 18.2% |
| 70–79 | 1,620,765 | 27.8% | 71,005 | 27.9% | 3,528 | 21.6% | 411 | 38.9% |
| ≥80 | 741,956 | 12.7% | 23,288 | 9.1% | 1,415 | 8.7% | 390 | 36.9% |
| Sex | | | | | | | | |
| Female | 522,738 | 9.0% | 27,475 | 10.8% | 1,694 | 10.4% | 24 | 2.3% |
| Male | 5,311,805 | 91.0% | 227,120 | 89.2% | 14,623 | 89.6% | 1,033 | 97.7% |
| Residence | | | | | | | | |
| Rural | 2,002,299 | 34.3% | 61,845 | 24.3% | 2,169 | 13.3% | 121 | 11.4% |
| Urban | 3,832,244 | 65.7% | 192,750 | 75.7% | 14,148 | 86.7% | 936 | 88.6% |
| Date of COVID-19 test | | | | | | | | |
| Feb 8–Apr 21 | n/a | n/a | 38,311 | 15.0% | 4,246 | 26.0% | 582 | 55.1% |
| Apr 22–Jun 21 | n/a | n/a | 120,313 | 47.3% | 4,379 | 26.8% | 349 | 33.0% |
| Jun 22–Jul 22 | n/a | n/a | 95,971 | 37.7% | 7,692 | 47.1% | 126 | 11.9% |
| Region | | | | | | | | |
| West | 1,162,263 | 19.9% | 56,303 | 22.1% | 2,863 | 17.5% | 128 | 12.1% |
| South | 2,697,047 | 46.2% | 113,594 | 44.6% | 8,324 | 51.0% | 371 | 35.1% |
| Midwest | 1,235,865 | 21.2% | 50,796 | 20.0% | 2,464 | 15.1% | 172 | 16.3% |
| Northeast | 737,135 | 12.6% | 33,902 | 13.3% | 2,666 | 16.3% | 386 | 36.5% |
| Outbreak pattern | | | | | | | | |
| Early | 795,442 | 13.6% | 44,786 | 17.6% | 4,453 | 27.3% | 523 | 49.5% |
| Late | 1,201,717 | 20.6% | 49,454 | 19.4% | 4,866 | 29.8% | 139 | 13.2% |
| Resurgent | 397,372 | 6.8% | 16,892 | 6.6% | 1,564 | 9.6% | 67 | 6.3% |
| Steady | 1,697,326 | 29.1% | 84,745 | 33.3% | 3,690 | 22.6% | 214 | 20.2% |
| Other | 1,742,686 | 29.9% | 58,718 | 23.1% | 1,744 | 10.7% | 114 | 10.8% |

Region based on US Census groupings. Outbreak pattern based on site-level percentage of positive tests per month among sites with at least 100 positive COVID-19 tests: early (≥10% in March or April), late (≥10% in June or July), resurgent (≥10% in March or April and June or July), steady (<10% in all months), or other (sites with <100 positive tests).

COVID-19, coronavirus disease 2019; n/a, not applicable.

were similar by race/ethnicity, and age ranged from 20 to 105 years, with 24% less than 50 years, 35% 50–69 years, 28% 70–79 years, and 13% ≥70 years of age. Of these, 254,595 (43.6 per 1,000 individuals in care) were tested for COVID-19, of whom 65% were White, 26% were Black, and 9% were Hispanic. There were 16,317 (6.4%) individuals who tested positive for COVID-19 between February 8 and July 22, 2020, of whom 44% were White, 40% were Black, and 16% were Hispanic. While 66% of all individuals in care resided in urban areas, 76% of those tested and 87% of those testing positive for COVID-19 resided in urban areas. The

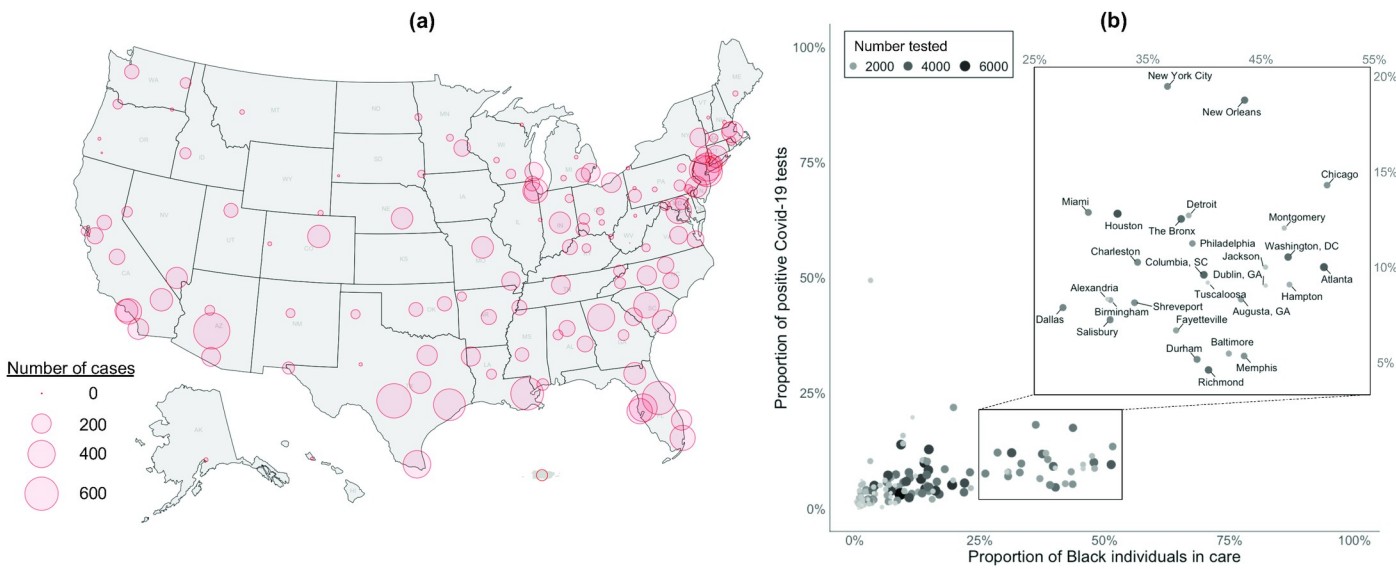

**Fig 1. Distribution of 16,317 laboratory-confirmed COVID-19 cases in the US Department of Veterans Affairs as of July 22, 2020.** (a) Distribution of all COVID-19 laboratory-confirmed cases in the US Department of Veterans Affairs between February 8 and July 22, 2020, included in the current study. (b) Proportion of positive COVID-19 test results by the proportion of Black individuals in care by site. Map created using R library USMAP (v0.5.0) and RStudio (v3.6.3). COVID-19, coronavirus disease 2019.

geographic distribution of COVID-19 cases in the VA was similar to the pattern of known hot-spots in the general population, including in the Northeast, South, and some Midwestern states (Fig 1). Several VA sites with the highest proportion of positive COVID-19 tests also performed a higher volume of tests and had the highest proportion of Black individuals in care, including New York City, New Orleans, and Chicago (Fig 1).

## Rate of testing and testing positive

Of the 254,595 patients tested for COVID-19, 73% received 1 test, 16% received 2 tests, 6% received 3 tests, and the remaining 5% received 4 or more tests. After reducing to 1 test per patient as described in Methods above, testing rates for COVID-19 were higher among Black (rate per 1,000 individuals: 60.0, 95% CI 59.6–60.5) and Hispanic individuals (52.7, 95% CI 52.1–53.4) compared to White individuals (38.6, 95% CI 38.4–38.7). Testing rates also varied by age, sex, rural/urban residence, region, and outbreak pattern (Table 2).

Among individuals tested for COVID-19, the proportion with a positive test varied by race/ethnicity (Table 2); 4.4% (95% CI 4.3%–4.5%) of White, 10.2% (95% CI 10.0%–10.4%) of Black, and 11.4% (95% CI 11.0%–11.9%) of Hispanic individuals tested positive for COVID-19. For White and Black individuals, the proportion of positive COVID-19 tests was highest at ages under 60 years and at or over 80 years. For Hispanic individuals, the proportion of positive test results was highest among lower age groups (15.6%, 95% CI 14.6%–16.6%, for 20–39 years) and continuously decreased with increasing age.

## Regression modeling of testing positive

Unadjusted associations between race/ethnicity and testing positive for COVID-19 yielded OR 2.49 (95% CI 2.40–2.58, $p < 0.001$) for Black and OR 2.80 (95% CI 2.67–2.94, $p < 0.001$) for Hispanic individuals compared to White individuals. After adjusting for age, the odds of testing positive did not change among Black individuals (OR 2.48, 95% CI 2.40–2.57, $p < 0.001$)

**Table 2. COVID-19 testing by race/ethnicity among all individuals in care as of July 22, 2020.**

| Characteristic | Testing rate per 1,000 (95% CI) | | | Percent testing positive (95% CI) | | |
|---|---|---|---|---|---|---|
| | White (n = 4,309,613) | Black (n = 1,089,883) | Hispanic (n = 435,047) | White (n = 166,213) | Black (n = 65,441) | Hispanic (n = 22,941) |
| All individuals | 38.6 (38.4–38.7) | 60.0 (59.6–60.5) | 52.7 (52.1–53.4) | 4.4 (4.3–4.5) | 10.2 (10.0–10.4) | 11.4 (11.0–11.9) |
| Age, years | | | | | | |
| 20–39 | 32.0 (31.5–32.4) | 39.5 (38.5–40.4) | 44.1 (42.9–45.2) | 6.1 (5.8–6.5) | 13.8 (13.0–14.7) | 15.6 (14.6–16.6) |
| 40–49 | 38.1 (37.5–38.7) | 45.9 (44.9–47.0) | 49.8 (48.2–51.6) | 4.8 (4.5–5.2) | 11.9 (11.1–12.7) | 13.6 (12.4–14.8) |
| 50–59 | 47.5 (46.9–48.0) | 60.4 (59.5–61.3) | 59.8 (58.0–61.7) | 4.6 (4.3–4.9) | 10.4 (10.0–11.0) | 11.2 (10.2–12.3) |
| 60–69 | 48.1 (47.7–48.6) | 71.1 (70.2–72.0) | 61.9 (60.2–63.6) | 3.7 (3.5–3.9) | 8.6 (8.2–9.0) | 9.1 (8.3–10) |
| 70–79 | 38.4 (38.1–38.7) | 67.9 (66.7–69.1) | 54.4 (52.8–56.0) | 3.7 (3.6–3.9) | 9.1 (8.6–9.7) | 8.1 (7.3–8.9) |
| ≥80 | 25.5 (25.1–25.9) | 69.7 (67.6–71.9) | 53.0 (50.6–55.4) | 4.9 (4.6–5.2) | 10.9 (10.0–12.0) | 6.8 (5.7–8.0) |
| Sex | | | | | | |
| Female | 49.9 (49.1–50.7) | 56.0 (54.9–57.1) | 57.7 (55.6–59.9) | 3.9 (3.6–4.2) | 9.0 (8.4–9.6) | 9.0 (7.9–10.2) |
| Male | 37.7 (37.5–37.9) | 60.8 (60.3–61.3) | 52.1 (51.4–52.8) | 4.3 (4.2–4.5) | 10.3 (10.0–10.5) | 11.5 (11.1–11.9) |
| Residence | | | | | | |
| Rural | 30.1 (29.9–30.4) | 39.8 (38.9–40.7) | 32.6 (31.3–34.0) | 2.6 (2.5–2.8) | 8.8 (8.1–9.5) | 7.9 (6.8–9.1) |
| Urban | 44.4 (44.1–44.6) | 64.0 (63.5–64.5) | 56.5 (55.7–57.2) | 5.1 (5.0–5.2) | 10.2 (10.0–10.5) | 11.5 (11.1–12.0) |
| Date of COVID-19 test | | | | | | |
| Feb 8–Apr 21 | n/a | n/a | n/a | 6.7 (6.4–7.0) | 19.4 (18.7–20.1) | 13.9 (12.7–15.0) |
| Apr 22–Jun 21 | n/a | n/a | n/a | 2.6 (2.5–2.7) | 5.8 (5.5–6.1) | 5.5 (5.1–6.0) |
| Jun 22–Jul 22 | n/a | n/a | n/a | 5.6 (5.4–5.8) | 11.2 (10.8–11.6) | 16.0 (15.2–16.7) |
| Region | | | | | | |
| West | 43.9 (43.5–44.4) | 72.9 (71.4–74.3) | 55.5 (54.4–56.7) | 4.2 (4.0–4.4) | 5.5 (5.1–6.0) | 9.0 (8.4–9.6) |
| South | 37.5 (37.2–37.8) | 51.9 (51.4–52.4) | 46.9 (46.1–47.8) | 4.7 (4.5–4.9) | 10.4 (10.1–10.7) | 12.4 (11.8–13.0) |
| Midwest | 36.4 (36.1–36.8) | 72.1 (70.8–73.4) | 52.4 (49.7–55.1) | 3.2 (3.0–3.4) | 10.5 (9.9–11.1) | 8.3 (6.9–9.8) |
| Northeast | 37.6 (37.1–38.1) | 89.6 (87.7–91.5) | 81.0 (78.1–83.9) | 5.3 (5.0–5.6) | 13.0 (12.3–13.8) | 14.6 (13.3–16.0) |
| Outbreak pattern | | | | | | |
| Early | 43.8 (43.3–44.4) | 84.4 (83.2–85.7) | 73.3 (71.1–75.6) | 6.4 (6.1–6.7) | 14.3 (13.8–14.9) | 13.5 (12.4–14.7) |
| Late | 37.2 (36.8–37.6) | 48.4 (47.6–49.2) | 50.0 (48.9–51.1) | 7.1 (6.8–7.4) | 11.4 (10.9–11.9) | 17.3 (16.5–18.2) |
| Resurgent | 34.4 (33.6–35.2) | 51.3 (50.3–52.4) | 50.2 (47.4–53.2) | 5.8 (5.2–6.3) | 11.4 (10.8–12.1) | 14.2 (12.2–16.4) |
| Steady | 45.5 (45.2–45.9) | 65.8 (65.0–66.7) | 57.4 (55.9–58.9) | 3.3 (3.2–3.5) | 6.4 (6.1–6.8) | 7.8 (7.1–8.5) |
| Other | 31.9 (31.6–32.2) | 47.0 (45.9–48.2) | 44.0 (42.8–45.1) | 2.5 (2.4–2.7) | 5.8 (5.2–6.4) | 3.5 (3.0–4.1) |

Region based on US Census groupings. Outbreak pattern based on site-level percentage of positive tests per month among sites with at least 100 positive COVID-19 tests: early (≥10% in March or April), late (≥10% in June or July), resurgent (≥10% in March or April and June or July), steady (<10% in all months), or other (sites with <100 positive tests).

CI, confidence interval; COVID-19, coronavirus disease 2019; n/a, not applicable.

and somewhat attenuated among Hispanic individuals (OR 2.56, 95% CI 2.44–2.69, $p < 0.001$) (Table 3). These associations further attenuated after additionally accounting for sex, rural/urban residence, and site of care among Black (OR 1.92, 95% CI 1.85–2.00, $p < 0.001$) and Hispanic individuals (OR 1.96, 95% CI 1.86–2.07, $p < 0.001$). These estimates were robust to any individual (S2 Table) or combined adjustment for comorbidities, substance use, and medication history. In fully adjusted models, Black (OR 1.93, 95% CI 1.85–2.01, $p < 0.001$) and Hispanic (OR 1.84, 95% CI 1.74–1.94, $p < 0.001$) individuals remained at increased odds of testing positive for COVID-19 (Fig 2).

The disparity in testing positive for COVID-19 between Black and White individuals decreased between the calendar periods February 8–April 21 (OR 2.16, 95% CI 1.98–2.36, $p < 0.001$) and June 22–July 22 (OR 1.74, 95% CI 1.64–1.85, $p < 0.001$) (Fig 3). By region, the

**Table 3. Associations with testing positive and subsequent 30-day mortality, February 8 to July 22, 2020.**

| Characteristic | Positive test result among tested (n/N = 16,317/254,595) | | | | | | 30-day mortality among cases* (n/N = 931/8,625) | | | | | |
|---|---|---|---|---|---|---|---|---|---|---|---|---|
| | Age-adjusted | | Demographic-adjusted[a] | | Fully adjusted[b] | | Age-adjusted | | Demographic-adjusted[a] | | Fully adjusted[b] | |
| | OR (95% CI) | p-Value | OR (95% CI) | p-Value | OR (95% CI) | p-Value | OR (95% CI) | p-Value | OR (95% CI) | p-Value | OR (95% CI) | p-Value |
| Race/ethnicity | | | | | | | | | | | | |
| White | ref | | ref | | ref | | ref | | ref | | ref | |
| Black | 2.48 (2.40–2.57) | <0.001 | 1.92 (1.85–2.00) | <0.001 | 1.93 (1.85–2.01) | <0.001 | 1.08 (0.93–1.26) | 0.32 | 1.02 (0.84–1.22) | 0.87 | 0.97 (0.80–1.17) | 0.74 |
| Hispanic | 2.56 (2.44–2.69) | <0.001 | 1.96 (1.86–2.07) | <0.001 | 1.84 (1.74–1.94) | <0.001 | 1.11 (0.85–1.44) | 0.46 | 0.98 (0.72–1.31) | 0.87 | 0.99 (0.73–1.34) | 0.94 |
| Age, years | | | | | | | | | | | | |
| 20–39 | 1.74 (1.65–1.83) | <0.001 | 1.75 (1.66–1.85) | <0.001 | 1.58 (1.49–1.68) | <0.001 | | | | | | |
| 40–49 | 1.45 (1.37–1.54) | <0.001 | 1.44 (1.36–1.53) | <0.001 | 1.28 (1.20–1.36) | <0.001 | | | | | | |
| 50–59 | 1.28 (1.22–1.35) | <0.001 | 1.27 (1.20–1.33) | <0.001 | 1.16 (1.10–1.22) | <0.001 | 0.18 (0.13–0.25)[c] | <0.001 | 0.20 (0.15–0.27)[c] | <0.001 | 0.27 (0.19–0.37)[c] | <0.001 |
| 60–69 | ref | | ref | | ref | | ref | | ref | | ref | |
| 70–79 | 0.87 (0.83–0.91) | <0.001 | 1.02 (0.97–1.07) | 0.41 | 0.93 (0.89–0.98) | 0.01 | 2.43 (2.00–2.95) | <0.001 | 2.34 (1.91–2.86) | <0.001 | 2.02 (1.64–2.48) | <0.001 |
| ≥80 | 1.08 (1.01–1.15) | 0.02 | 1.28 (1.20–1.36) | <0.001 | 1.08 (1.01–1.16) | 0.02 | 6.39 (5.21–7.85) | <0.001 | 5.92 (4.78–7.34) | <0.001 | 4.59 (3.64–5.78) | <0.001 |
| Sex, male versus female | 1.27 (1.20–1.34) | <0.001 | 1.38 (1.31–1.46) | <0.001 | 1.58 (1.49–1.67) | <0.001 | 1.59 (1.00–2.53) | 0.048 | 1.53 (0.96–2.45) | 0.07 | 1.48 (0.92–2.36) | 0.11 |
| Residence, urban versus rural | 2.10 (2.00–2.20) | <0.001 | 1.39 (1.32–1.46) | <0.001 | 1.39 (1.32–1.46) | <0.001 | 1.16 (0.92–1.47) | 0.22 | 1.06 (0.81–1.39) | 0.68 | 1.03 (0.79–1.36) | 0.81 |

*Models of 30-day mortality limited to cases testing positive for COVID-19 on or before June 21, 2020, to allow 30 days of follow-up.

[a]Additionally adjusted for sex and rural/urban residence, and conditioned on site of care.

[b]Additionally adjusted for baseline comorbidity (asthma, cancer, chronic kidney disease, chronic obstructive pulmonary disease, diabetes mellitus, hypertension, liver disease, vascular disease), substance use (alcohol consumption, alcohol use disorder, smoking status), and medication history (angiotensin converting enzyme inhibitor, angiotensin II receptor blocker).

[c]OR for age 20–59 years. Low number of mortality events in age groups 20–39 and 40–49 years, thus grouped with age group 50–59 years.

CI, confidence interval; COVID-19, coronavirus disease 2019; OR, odds ratio.

disparity between Black and White individuals was highest in the Midwest (OR 2.66, 95% CI 2.41–2.95, $p < 0.001$) than any other region, and lowest in the West (OR 1.24, 95% CI 1.11–1.39, $p < 0.001$). By outbreak pattern, the disparity between Black and White individuals was highest at VA sites that experienced an early (OR 2.11, 95% CI 1.95–2.28, $p < 0.001$) or resurgent (OR 2.06, 95% CI 1.81–2.35, $p < 0.001$) outbreak and lowest at VA sites that experienced a late outbreak (OR 1.66, 95% CI 1.54–1.80, $p < 0.001$). In contrast, there was no variation observed in the disparity between Hispanic and White individuals by calendar time, region, or outbreak pattern.

## Regression modeling of 30-day mortality

There were 8,625 individuals who tested positive for COVID-19 on or before June 21, 2020, of whom 931 (457 [49%] White; 392 [42%] Black; and 82 [9%] Hispanic) died within 30 days. Unadjusted associations between race/ethnicity and mortality within 30 days of a positive test yielded OR 0.76 (95% CI 0.66–0.88, $p < 0.001$) for Black and OR 0.60 (95% CI 0.47–0.77, $p <$

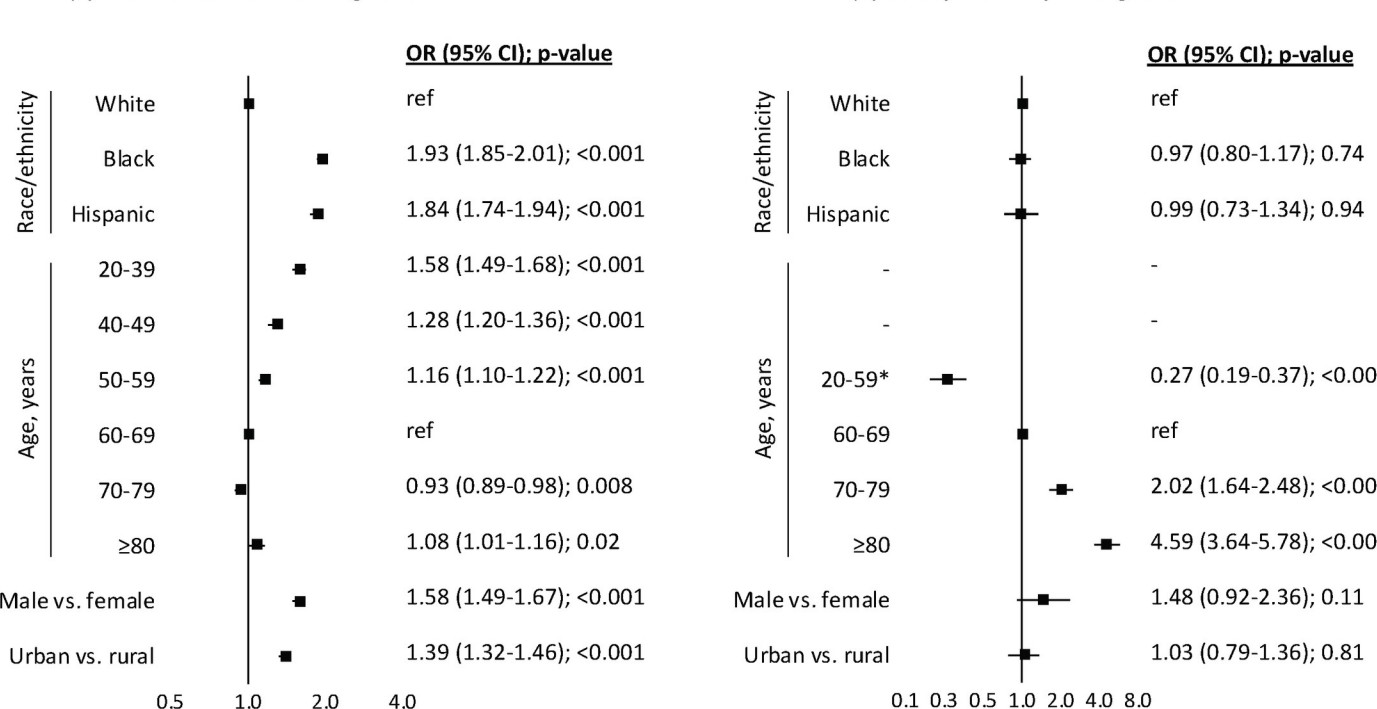

**Fig 2. Adjusted associations of demographic characteristics with testing positive for COVID-19 and subsequent 30-day mortality as of July 22, 2020.** (a) Positive test result among tested; (b) 30-day mortality among cases. Both models were conditioned on site of care and adjusted for baseline comorbidity (asthma, cancer, chronic kidney disease, chronic obstructive pulmonary disease, diabetes mellitus, hypertension, liver disease, vascular disease), substance use (alcohol consumption, alcohol use disorder, smoking status), and medication history (angiotensin converting enzyme inhibitor, angiotensin II receptor blocker). *Low number of mortality events in age groups 20–39 and 40–49 thus grouped with 50–59. CI, confidence interval; COVID-19, coronavirus disease 2019; OR, odds ratio.

0.001) for Hispanic individuals compared to White individuals. This association was not observed after adjusting for age among Black (OR 1.08, 95% CI 0.93–1.26, $p = 0.32$) and Hispanic individuals (OR 1.11, 95% CI 0.85–1.44, $p = 0.46$) (Table 3). The null association was robust to any further adjustment (Fig 2). Most deaths ($n = 603$, 65%) occurred in hospital. In sensitivity analyses, results from a model of 30-day mortality restricted to inpatient deaths did not alter the conclusions from the primary analyses (OR 1.12, 95% CI 0.90–1.40, $p = 0.32$, among Black individuals; OR 1.08, 95% CI 0.76–1.54, $p = 0.65$, among Hispanic individuals).

## Discussion

This study examined racial and ethnic disparities in testing and subsequent COVID-19 mortality among approximately 6 million individuals receiving care in the US. We found that Black and Hispanic individuals were more likely to be tested and to test positive for COVID-19 than White individuals, even after comprehensive adjustment for underlying health conditions, other demographics, and geographic location. Among the variables assessed in this study, age, rural/urban residence, and site of care explained more of the racial/ethnic disparity in testing positive for COVID-19 than comorbidities, substance use, or medication history. While the disparity between Black and White individuals decreased over time, the disparity was strongest in the Midwest and at VA sites that experienced an early or resurgent outbreak. There was no variation observed in the disparity between Hispanic and White individuals by calendar time, region, or outbreak pattern. While individuals from minority backgrounds

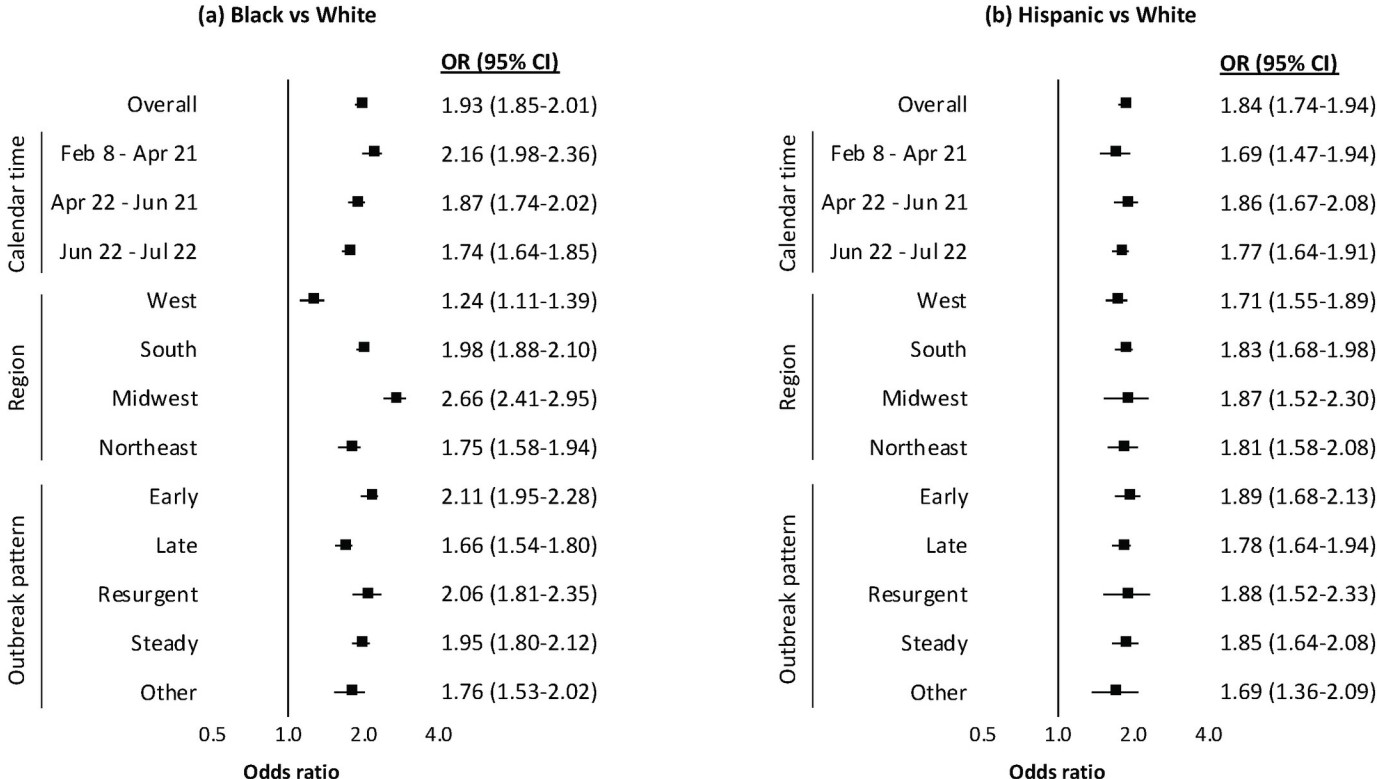

**Fig 3. Racial and ethnic disparities in testing positive for COVID-19, by calendar time, region, and outbreak pattern.** (a) Black versus White individuals; (b) Hispanic versus White individuals. All $p < 0.001$. Region based on US Census groupings. Outbreak pattern based on site-level percentage of positive tests per month among sites with at least 100 positive COVID-19 tests: early ($\geq$10% in March or April), late ($\geq$10% in June or July), resurgent ($\geq$10% in March or April and June or July), steady (<10% in all months), other (sites with <100 positive tests). Models were conditioned on site of care and adjusted for baseline comorbidity (asthma, cancer, chronic kidney disease, chronic obstructive pulmonary disease, diabetes mellitus, hypertension, liver disease, vascular disease), substance use (alcohol consumption, alcohol use disorder, smoking status), and medication history (angiotensin converting enzyme inhibitor, angiotensin II receptor blocker). CI, confidence interval; COVID-19, coronavirus disease 2019; OR, odds ratio.

appeared to experience excess burden of COVID-19, among those infected, there was no observed difference in 30-day mortality by race/ethnicity group. The apparent racial/ethnic disparity in mortality in unadjusted data was principally explained by differing age structures between the populations.

## Key strengths and limitations

This study elucidated racial and ethnic disparities in testing patterns of COVID-19 independent of underlying health status and other key factors in a nationwide sample. Strengths of this study included that it was based on well-annotated electronic health record data from a team with decades of experience using VA data, enabling a rapid and reliable analysis of COVID-19 outcomes by race and ethnicity. This analysis utilized patients' records from an entire healthcare system, which made it less prone to collider bias (i.e., non-random selection of individuals into a study) than other COVID-19 studies limited to individuals testing positive or admitted to hospital [21]. Unlike other nationwide healthcare systems, linkage to COVID-19 testing data or outcomes was not required as the integrated nature of VA healthcare provided at over 1,200 sites allows all information to be stored in its Corporate Data Warehouse. We used validated algorithms to accurately extract information on and adjust models for a wide range of clinical, behavioral, and geographic factors, with very little missingness in the data. The scale

of VA data also allowed us to assess the impact of COVID-19 separately across multiple racial and ethnic minority groups; combining or limiting analyses to a single minority group would have masked important differences between Black and Hispanic individuals. We continue to monitor COVID-19 outcomes for individuals of other minority backgrounds and plan to follow up these analyses when there are sufficient numbers for analysis.

While this analysis adds information, its limitations must be kept in mind. First, this study was conducted on veterans currently receiving care in the VA, who are older and have a higher prevalence of chronic health conditions and risk behaviors than the general US population [22–24]. However, prior research has established that after adjusting for age, sex, race/ethnicity, region, and rural/urban residence, all of which were included in this study, there is no difference in total disease burden between veterans and non-veterans [24]. Our key finding of no observed racial disparity of COVID-19 mortality has also been shown in a smaller non-veteran population [9]; thus, associations reported in this study are likely generalizable to the wider US population. Second, while individuals in VA care represent a diversity of backgrounds, women represented a small proportion of individuals in the sample. Thus, our analysis was not powered to assess interactions between sex and race/ethnicity. Third, beyond adjusting for rural/urban location and site of testing, we were not able to explore likely social determinants of the pronounced differential burden of COVID-19 among minority individuals. More detailed information on nursing home residence and socioeconomic status (e.g., type of employment, income, number of individuals in household) were unavailable or not consistently recorded in VA data, as is the case in most other electronic health record data sources. Fourth, as is true outside the VA, only a small proportion of individuals have been tested (~5%), and rates of testing vary by site and within important subgroups. However, while initial testing was limited, by mid-April the VA began testing all individuals admitted to hospital and before any inpatient or outpatient procedures, even in those not suspected to have COVID-19. Our models for testing positive should be cautiously interpreted as a proxy of odds of infection since those with mild symptoms were unlikely to have received testing, particularly in the early stages of the outbreak.

## Findings in context

Our findings of racial and ethnic disparities in COVID-19 provide important distinctions from previous reports in the US and other countries with ethnically diverse populations. To our knowledge, one of the largest studies to date on racial disparities in COVID-19 outcomes in the US followed 3,481 COVID-19 cases in the state of Louisiana and found that non-Hispanic Black individuals represented 77% of hospitalizations and 71% of deaths despite only making up 31% of the total source population [9]. However, this study was based on patients who tested positive for COVID-19 in a statewide healthcare system and was underpowered to investigate ethnic minorities. We were able to expand the scope of this finding nationally and to include Hispanic individuals. In the UK, which was the first country with a broadly ethnically diverse population to experience a COVID-19 outbreak [25], a study of 17 million individuals showed that those from minority backgrounds had a substantially higher risk of mortality from COVID-19, which was not fully explained by underlying health conditions or social deprivation [26]. While our study also found racial and ethnic disparities, we found that these disparities occurred primarily at a stage prior to hospitalization (i.e., testing positive for COVID-19). We found no evidence of racial or ethnic disparities in 30-day mortality once models were restricted to those who tested positive for COVID-19. Our findings may be an underestimate of the US population risk as health disparities in the VA tend to be smaller than in the private sector [27]. Nevertheless, at a population level the substantial excess burden of

SARS-CoV-2 infection among Black and Hispanic individuals inevitably translates to excess COVID-19 mortality in these communities.

We demonstrated that Black and Hispanic individuals were more likely to test positive than their White counterparts even after accounting for underlying health conditions, other demographics, rural/urban residence, and site of care. Based on experience with the 1918 Spanish flu and the 2009 H1N1 epidemic, public health experts have warned that racial and ethnic minority populations may be at higher risk during infectious disease outbreaks due to underlying health conditions, lower access to care, and socioeconomic conditions [28,29]. Notably, our analysis found that underlying health conditions did not explain any of the disparity between racial/ethnic groups in the odds of testing positive for COVID-19 or subsequent mortality in models already accounting for demographics, principally age, rural/urban residence, and VA site of care—essential information to help guide effective interventions. Prior reports have also highlighted that members of racial and ethnic minorities are more likely to live in densely populated areas or multigenerational households, and minority groups are overrepresented in jails, prisons, and detention centers, all of which lead to reduced capacity to implement physical distancing [30–34]. Similarly, Black and Hispanic workers are more likely than their White counterparts to be workers in essential industries, who continue to work outside the home despite outbreaks in their communities, making them more prone to exposure and therefore infection [34–36].

We found substantial variation in the disparity between Black and White individuals in testing positive for COVID-19 by geographic region, with stronger disparity observed in the Midwest than all other regions, and disparity most attenuated in the West. Further breakdown of groups within the Black community (e.g., African American, Afro-Caribbean, African), which could potentially reveal additional variation, is not captured in VA data. The observed disparities may be due to differential social determinants of health between Black and White individuals across regions. A US Census Bureau report showed that while racial residential segregation has diminished over time nationally, communities in the Midwest remained less integrated than in the West [37]. If community-level exposure is driving risk of SARS-CoV-2 infections, then the disparity in testing positive for COVID-19 may be lower in regions with greater integration between White and Black residents, as is the case in the West. We also found that the disparity between Black and White individuals in testing positive slightly decreased over the study period and was highest at VA sites that experienced an early outbreak of COVID-19. This finding may be partially explained by the increased attention on racial disparities in COVID-19 in the media [3–5] that may have impacted behaviors like wearing face coverings in public to reduce the spread of infection [38].

Interestingly, the ethnic disparity between Hispanic and White individuals in testing positive for COVID-19 was consistent across time, geographic region, and outbreak pattern; the disparity was consistently observed across all strata. The lack of variation over time may be explained, in part, by less nationwide media coverage and epidemiological investigations of outbreaks of COVID-19 in Hispanic communities. Importantly, the Hispanic population in the US comprises a wide array of ethnic communities (e.g., Mexican, Puerto Rican, Cuban). However, these distinctions are not captured in VA data. The umbrella grouping may mask any potential variation within such a heterogeneous population. Further research on the impact of COVID-19 in Hispanic and Latinx communities is urgently needed.

Testing rates for COVID-19 in the VA were higher among Black and Hispanic individuals compared to White individuals. Local reporting from metropolitan areas with large minority populations, including New York City [39] and Chicago [40], has highlighted the disproportionate impact of COVID-19 in minority communities. We showed that VA facilities in these cities and others around the country that conducted the highest number of COVID-19 tests

also had the highest proportion of Black individuals in care. There were also differences in the rate of COVID-19 testing and the proportion testing positive by age, sex, and type of residence across race/ethnicity groups. These findings demonstrate the need for epidemiological investigations to characterize testing patterns in the underlying population as they provide important context for interpretations of models of COVID-19 outcomes. To our knowledge, the largest medical record study to date analyzed COVID-19-related mortality in a population of 17 million residents in the UK, the vast majority of whom never tested for COVID-19 [26]. While the authors identified ethnic disparities in COVID-19-related mortality, the estimates reported can be interpreted as the overall burden of mortality by ethnicity without accounting for underlying testing patterns. We found a similar disparity in the overall burden of COVID-19-related death in the full source population of approximately 6 million individuals in care at the VA. However, when the model was restricted to individuals testing positive—which inherently accounts for factors related to access to testing, non-random testing, and odds of infection—racial and ethnic disparities in mortality were no longer observed.

## Policy implications

These findings underscore the urgent need to proactively tailor strategies to contain and prevent further outbreaks in the US, principally focused on testing and getting individuals into care. Black and Hispanic communities are at increased risk of infection, justifying increased intensity of intervention. Our findings of variation in disparities over time and across geographic regions highlight the important need for community-based interventions at a state and local level to contain further exposure and outbreaks of COVID-19, particularly tailored to minority communities. Other interventions may include clinical decision support tools to prompt educational and testing interventions based upon an individualized risk assessment of testing positive for COVID-19. Outreach products about COVID-19 testing and disparities should also be distributed to patient advocates and groups at all points of care.

## Future research

We appeal to other researchers investigating racial and ethnic disparities to perform analyses on the entire population at risk for COVID-19 where data are available, and to compare findings associated at each stage in the clinical course of COVID-19, from testing to outcomes. In this paper, we focused only on 30-day mortality among COVID-19 cases. We plan to explore other outcomes, including hospitalization, intensive care, and intubation, in subsequent analyses to examine whether racial and ethnic disparities exist in the clinical course of COVID-19 after testing positive and before death. Among other factors, future research should consider the role of other social determinants of health, including employment type, number of individuals in household, nursing home residence, and incarceration. Other racial and ethnic minorities in the US deserve attention, and while we did not have enough statistical power to include other groups in this analysis, we will continue to monitor these numbers for future research.

## Supporting information

**S1 STROBE Checklist. STROBE statement checklist.**
(DOCX)

**S1 Table. International Classification of Diseases–10th Revision, Clinical Modification (ICD-10-CM) diagnosis codes.**
(XLSX)

**S2 Table. Individual adjustments for the association between race/ethnicity and COVID-19 positivity and mortality.**
(DOCX)

## Acknowledgments

The authors wish to recognize Dr. Kendall Bryant as the NIAAA Scientific Collaborator for this study.

The views and opinions expressed in this paper are those of the authors and do not necessarily represent those of the Department of Veterans Affairs or the United States Government.

## AcknowledgmentsTransparency declaration

CTR and ACJ affirm that this paper is an honest, accurate, and transparent account of the study being reported; that no important aspects of the study have been omitted; and that any discrepancies from the study as planned have been explained.

## Author Contributions

**Conceptualization:** Christopher T. Rentsch, Amy C. Justice.

**Data curation:** Christopher T. Rentsch, Farah Kidwai-Khan, Janet P. Tate, Lesley S. Park, Joseph T. King Jr, Melissa Skanderson, Ronald G. Hauser, Anna Schultze, Mark Holodniy.

**Formal analysis:** Christopher T. Rentsch.

**Funding acquisition:** Amy C. Justice.

**Methodology:** Christopher T. Rentsch, Janet P. Tate, Lesley S. Park, Ronald G. Hauser, Vincent Lo Re III, Kathleen M. Akgün, Kristina Crothers, Matthew S. Freiberg, Amy C. Justice.

**Project administration:** Christopher T. Rentsch, Amy C. Justice.

**Visualization:** Christopher T. Rentsch, Janet P. Tate, Lesley S. Park, Christopher I. Jarvis.

**Writing – original draft:** Christopher T. Rentsch, Amy C. Justice.

**Writing – review & editing:** Christopher T. Rentsch, Farah Kidwai-Khan, Janet P. Tate, Lesley S. Park, Joseph T. King Jr, Melissa Skanderson, Ronald G. Hauser, Anna Schultze, Christopher I. Jarvis, Mark Holodniy, Vincent Lo Re III, Kathleen M. Akgün, Kristina Crothers, Tamar H. Taddei, Matthew S. Freiberg, Amy C. Justice.

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
