## [Editor Report · Decision Letter 0]

12 Jun 2020

Dear Dr Rentsch, 

Thank you for submitting your manuscript entitled "Covid-19 by Race and Ethnicity: A National Cohort Study of 6 Million United States Veterans" for consideration by PLOS Medicine.

Your manuscript has now been evaluated by the PLOS Medicine editorial staff and I am writing to let you know that we would like to send your submission out for external assessment.

Kind regards,

Richard Turner, PhD

Senior editor, PLOS Medicine

rturner@plos.org

---

## [Decision Letter · Decision Letter 1]

20 Jul 2020

Dear Dr. Rentsch,

Thank you very much for submitting your manuscript "Covid-19 by Race and Ethnicity: A National Cohort Study of 6 Million United States Veterans" (PMEDICINE-D-20-02701R1) for consideration at PLOS Medicine. 

Your paper was discussed among the editorial team and and sent to independent reviewers, including a statistical reviewer. The reviews are appended at the bottom of this email and any accompanying reviewer attachments can be seen via the link below:

[LINK]

In light of these reviews, we will not be able to accept the manuscript for publication in the journal in its current form, but we would like to invite you to submit a revised version that fully addresses the reviewers' and editors' comments. You will appreciate that we cannot make a decision about publication until we have seen the revised manuscript and your response, and we expect to seek re-review by one or more of the reviewers. 

We hope to receive your revised manuscript by Aug 03 2020 11:59PM. Please email us (plosmedicine@plos.org) if you have any questions or concerns.

Please let me know if you have any questions. Otherwise, we look forward to receiving your revised manuscript soon. 

Sincerely,

Richard Turner, PhD

rturner@plos.org

In your data statement, please explain briefly the reasons for non-availability of study data, e.g., ethics criteria. 

Please revisit your title, so that it is a better match with journal style. We suggest "Patterns of COVID-19 testing and mortality by race and ethnicity in United States veterans: a national cohort study". 

Please add a new final sentence to the "methods and findings" subsection of your abstract, beginning "Study limitations include ...", or similar, and quoting 2-3 of the study's main limitations. 

At line 72, please begin the sentence "In this study, we found that ..." or similar. 

After the abstract, please add a new and accessible "author summary" section in non-identical prose. You may find it helpful to consult one or two recent research papers published in PLOS Medicine to get a sense of the preferred style. 

Early in the methods section of your main text, please state whether the study had a protocol or prespecified analysis plan, and if so attach the document as a supplementary file (referred to in the text). Please highlight analyses that were not prespecified. 

Please remove the "role of the funding source" statement from the main text. Funding and competing interest information will appear in the article metadata in the event of publication, via information provided in the submission form. 

Where available, please add p values along with odds ratios and 95% CI.

Please avoid claims of "the first" or "largest", e.g., at line 251, and where these cannot be avoided add "to our knowledge" or similar. 

Again, please remove information on data sharing, funding and competing interests from the end of the main text, as this belongs in the metadata.

We suggest using initial capitals for "white" and "black" throughout your text.

Please add full access details to items in your reference list where needed, e.g. to reference 30. 

Where you cite preprints, such as reference 2, please add "[preprint]".

Please add a completed checklist for the most appropriate reporting guideline - which may be STROBE or RECORD - as a supplementary document, referred to early in your methods section. In the checklist, please refer to individual items by section (e.g., "Methods") and paragraph number rather than by page or line numbers, as the latter generally change in the event of publication, 

Comments from the reviewers:

*** Reviewer #1: 

"Covid-19 by Race and Ethnicity: A National Cohort Study of 6 Million United States Veterans" describes the results of a detailed multivariate analysis using data from the largest integrated healthcare system in the U.S., the Department of Veterans Affairs (VA). The primary conclusions were that minority (i.e. Black & Hispanic) individuals experienced an excess burden of Covid-19 infection, as indicated by significantly higher adjusted OR of testing positive for Covid-19, compared to whites.

As emphasized in the manuscript, a key strength of this study is the scale and completeness of the data used, which covers over 5.8 million individuals from a single healthcare system. The main statistical analysis applied on Covid-19 positive individuals (summarized in Table 3) also appears to be with relatively standard logistic regression models.

However, a major limitation would be the relatively small number of non-medical input variables available (age, race/ethnicity, sex, residence urban/rural). As such, it is not possible to quantify whether the difference in burden might be explained to some extent by socioeconomic factors, which seems not improbable (e.g. access to private testing). Nevertheless, the unprecedented scope of this study as regards the distribution of Covid-19 burden by race/ethnicity should make it an important addition to the literature, especially if the limitations pertaining to available factors are clearly acknowledged upfront.

Nevertheless, there remain some possible points for consideration:

1. The adjustments made for various confounders (e.g. comorbidities, stations) in the various statistical analyses as reported in Table 3/Table S2 are not fully described. The authors might consider releasing the code used, and/or detail the adjustment procedure in the supplementary material (particularly for the various individual comorbidities), although this is not expected to change the conclusions materially.

2. The tests per 1000 individuals metric (as seen in Line 161/Table 2) might be clarified. Does it refer to raw number of tests (i.e. an individual might have multiple tests applied, as mentioned in Line 128), or does it instead refer to the number of tested individuals per 1000 individuals? In either case, the prevalence of multiple tests per individual (also broken down by race/ethnicity if possible) might be briefly described.

3. From Lines 110/130, it appears that data may be available as to the location of testing (i.e. hospitalization); the authors might consider reporting distribution of broad location types (e.g. primary care, urgent care, emergency department, inpatient, from [8]) and as a possible confounder, if appropriate.

4. From Table 3, the number of cases/events decreases as additional confounders are considered. It is assumed that this is due to missing data for these additional confounders (as suggested for smoking/alcohol consumption in Line 156); the authors might consider explicitly stating this, if true.

5. From Table 3, The large difference in OR for Black individuals, between the Multivariable model and the Conditional OR model (2.73 vs. 1.96), might merit a brief comment on its possible significance.

*** Reviewer #2: 

This paper is a strong analysis overall of a large and important dataset. I have a few specific questions that I think should be addressed before the paper is acceptable for publication. The inclusion of data on who did and did not get tested is of particular value, as most studies of this nature have focused only on positive test results.

1. The authors claim that the finding that there is no racial disparity in mortality among individuals in the VA system after adjustment for comorbidities is not supported by the data if the authors intend for it to be interpreted as a population-level finding. It may be true that these disparities are minimized in a population with access to quality care, but the discussion makes it appear as if these findings are generalizable to the total population, which they are not. 

2. More information on regional or site-specific variation in these effects would strengthen the results. Because the pandemic is a non-stationary event, the different VA centers included may have been in different phases of pandemic response within the individual time-slices analyzed here. At the very least, presenting results regionally would allow the reader to see to what extent these results vary by location. Because the dataset is so large, there is no reason that the analysis cannot just be repeated on region-specific subsets of the data. 

3. The 30-day mortality analyzed by the authors is essentially the same as estimating the case-fatality rate. Although 30-day mortality is a standard term/measure, the authors should at least explain the relationship of this metric to the CFR to ensure that the analysis is accessible to a wide audience.

*** Reviewer #3: 

This study provides a simple but geographically expansive approach to evaluating underlying COVID-19 testing patterns and race/ethnic disparities. This is an important and convincing study overall, but I do have a number of comments and concerns:

Line 87—suggest providing an example of incidence disparities to motivate this prioritization statement

Line 135—please clarify if you looked at 30-day mortality from all-causes or from COVID-19 only. If the former, this requires additional clarification in terms of the leading causes of death and COVID-19 role. If the latter, how did you deal with competing risks? On what basis was 30 days chosen?

Line 148—please clarify how comorbidities of interest were selected (which evidence base). Also, were these all modeled individually? There are many covariates and these models risk overadjustment, particularly given that some (e.g. asthma) are only tenuously linked to more severe COVID-19 outcomes.

Line 220—there was not an 'increased likelihood' (a measure of probability) of testing positive, but instead increased odds. Please check manuscript throughout for similar errors.

Line 224—site of care is not properly defined previously

Lines 226-229—this is an erroneous interpretation and runs into Table 2 fallacy issues—the analytic hypothesis examines race and testing, and thus cannot inform a conclusion regarding gender and urban status in the multivariate analyses (descriptive interpretations only are appropriate). Displaying the adjustment variable parameter estimates (and interpreting them) is inappropriate because they may be confounded by additional variables that were not included. See e.g. https://pubmed.ncbi.nlm.nih.gov/23371353/?from_term=table+2+fallacy&from_pos=1

Lines 237-238: The sentence "There was some suggestion of an association between male sex (OR 1.44, 95% CI 0.75-2.76) or urban residence (OR 1.57, 95% CI 0.96-2.57) and 30-day mortality, but confidence intervals were wide" can be removed. There is no suggestion of an association given these wide confidence intervals. The results just as likely imply a lack of an association (especially for male sex) as they do an association.

Discussion section: The authors make no attempt at interpreting the change in trends of time for racial group testing results. Without additional clarification, it is unclear what these results might mean.

Policy implications: What about household-level interventions to contain spread, or interventions at the local/state level to enforce quarantine measures etc.?

***

[LINK]

---

## [Decision Letter · Decision Letter 2]

14 Aug 2020

Dear Dr. Rentsch,

Thank you very much for re-submitting your manuscript "Patterns of Covid-19 testing and mortality by race and ethnicity among United States Veterans: a national cohort study" (PMEDICINE-D-20-02701R2) for consideration at PLOS Medicine.

I have discussed the paper with editorial colleagues and it was also seen again by one reviewer. I am pleased to tell you that, provided the remaining editorial and production issues are dealt with, we expect to be able to accept the paper for publication in the journal.

[LINK]

Please let me know if you have any questions. Otherwise, we look forward to receiving the revised manuscript soon. 

Sincerely,

Richard Turner, PhD

rturner@plos.org

Requests from Editors:

According to PLOS' data policy, https://journals.plos.org/plosmedicine/s/data-availability, we will need to ask you to include a non-author or institutional contact for readers interested in inquiring about access to study data. 

Please substitute "nationwide" for "national" in the title and where this occurs elsewhere in the article. 

Please make that "COVID-19" throughout the ms. 

At line 76, we suggest amending the text to "... under-representation of women and a lack of detailed information on social determinants ...". 

At line 91, please make that "... has yet investigated".

At line 318, do you mean "The absence of ethnic/racial disparity in mortality ..."? Alternatively, perhaps you wish to refer to the apparent disparity in unadjusted data, suggesting populations' differing age structures as a possible explanation for perceived disparities. 

Where "p<0.0001" is quoted, in the abstract and elsewhere, please substitute exact p values or "p<0.001".

Please avoid "p<0.000", e.g., in table 3.

Please add "[preprint]" to reference 16.

Comments from Reviewers:

*** Reviewer #1: 

We thank the authors for largely addressing the previously-raised concerns in the revised and expanded manuscript. In particular, additional details on the adjustments have been added to the text and in supplementary material (Table S2, which also explores the incremental effect of various confounders on OR), the tests per 1000 individuals metric has been clarified as testing rate, and no exclusions have now been implemented for Table 3.

***

[LINK]

---

## [Editor Report · Decision Letter 3]

31 Aug 2020

Dear Dr. Rentsch, 

On behalf of my colleagues and the academic editor, Dr. Jonathan Zelner, I am delighted to inform you that your manuscript entitled "Patterns of COVID-19 testing and mortality by race and ethnicity among United States Veterans: a nationwide cohort study" (PMEDICINE-D-20-02701R3) has been accepted for publication in PLOS Medicine. 

PRODUCTION PROCESS

PRESS

PROFILE INFORMATION

Thank you again for submitting the manuscript to PLOS Medicine. We look forward to publishing it. 

Best wishes, 

Richard Turner, PhD

Senior Editor 

PLOS Medicine

plosmedicine.org